# The Prognostic Value of Combined Status of Body Mass Index and Psychological Well-Being for the Estimation of All-Cause and CVD Mortality Risk: Results from a Long-Term Cohort Study in Lithuania

**DOI:** 10.3390/medicina58111591

**Published:** 2022-11-03

**Authors:** Dalia Lukšienė, Abdonas Tamosiunas, Ricardas Radisauskas, Martin Bobak

**Affiliations:** 1Department of Population Studies of Institute of Cardiology, Medical Academy, Lithuanian University of Health Sciences, LT-50162 Kaunas, Lithuania; 2Department of Environmental and Occupational Medicine, Faculty of Public Health, Medical Academy, Lithuanian University of Health Sciences, LT-47181 Kaunas, Lithuania; 3Department of Epidemiology and Public Health, University College London, London WC1E 6BT, UK

**Keywords:** body mass index, psychological well-being, all-cause mortality risk, cardiovascular diseases

## Abstract

*Background and Objectives:* It is very important to analyze how body mass index (BMI) and psychological well-being (PWB) combination may be differentially associated with mortality risk. The aim of this study was to evaluate the additional prognostic value of the combined status of BMI and PWB for the estimation of all-cause and cardiovascular disease (CVD) mortality risk in the adult Lithuanian urban population. *Materials and Methods:* Initial data were collected within the framework of the international cohort HAPIEE study from 2006 to 2008. A random sample of 7115 individuals aged 45–72 years was screened. The response rate was 65%. Deaths were evaluated by the death register of Kaunas city (Lithuania) in a follow-up study until 31 December 2020. The mean (SD) duration of the follow-up for the endpoints period was 12.60 (2.79) years. PWB was evaluated by a CASP-12 questionnaire. *Results:* The findings from the Cox proportional hazards regression multivariable analysis showed that the combinations of underweight plus lower PWB and severe obesity plus lower PWB increased all-cause mortality risk in men (respectively hazard ratio (HR) = 5.65 and HR = 1.60) and in women (respectively HR = 6.02 and HR = 1.77); and increased the risk of mortality from CVD in men (respectively HR = 6.69 and HR = 2.19) compared with responders with normal weight plus higher PWB. The combination of severe obesity plus higher PWB significantly increased the risk of all-cause and CVD mortality risk in men. The combinations of normal weight plus lower PWB and overweight plus lower PWB significantly increased the risk of all-cause mortality risk in men. *Conclusions:* The combination of severe obesity independently on lower or higher PWB and the combination of underweight plus lower PWB is a strong predictor for all-cause and CVD mortality risk in men and a strong predictor for all-cause mortality risk in women.

## 1. Introduction

For at least four decades, the prevalence of underweight has decreased, and that of obesity has increased, in most countries with significant variation in the magnitude of these changes across regions of the world [1,2,3]. From 1985 to 2016, age-standardized mean body mass index (BMI) increased by 1–4 kg/m^2^ in all regions, except for women in the high-income Asia Pacific region and Central and Eastern Europe whose mean BMI changed by less than 1 kg/m^2^ [3]. Numerous studies have shown the relationships between BMI status and an increased risk of all-cause mortality and mortality from cardiovascular diseases (CVD) [4,5,6,7,8]. Lower psychological well-being (PWB) has been identified as one of the leading risk factors for noncommunicable diseases [9,10,11,12,13]. Many studies analyzed these two risk factors separately [4,5,6,7,8,9,10]. The Encyclopedia of Gerontology and Population Aging emphasizes that obesity can affect emotional well-being through psychological, physical, and social factors related to social acceptance and self-perception [14]. Indeed, the scientific results show that being obese is frequently related to deteriorated psychological well-being (PWB) and quality of life, in comparison with normal-weight individuals [15,16]. Results of some studies indicated that being obese, or overweight is often associated with impaired quality of life and PWB in comparison with normal-weight people, both in developed and developing countries [14,15,17,18].

In Lithuania, the incidence and mortality rates of non-communicable diseases, especially CVD, are higher than in most European countries, especially when compared to high-income Western European countries [19]. Lithuania is characterized as the most prevalent obese people country in the European Union: one in six adults were obese (18%) in 2019, compared to 16% on average across the EU [20]. Thus, overweight, obesity, and PWB may be differentially associated with the risk of mortality in Lithuania. We hypothesized that increased BMI level and low PWB are strongly positively associated with the risk of mortality and have multiplicative interaction. The aim of this study was to evaluate the additional prognostic value of the combined status of BMI and PWB for the estimation of all-cause and CVD mortality risk in the middle age and elderly population.

## 2. Materials and Methods

### 2.1. Study Design

A prospective cohort study was carried out in the framework of the international Health, Alcohol and Psychosocial Factors in Eastern Europe (HAPIEE) study conducted in Kaunas (Lithuania) [21]. The baseline survey of this study took place in 2006 to 2008. Standardized and validated questionnaires and standard measuring methods for determining lifestyle, psychologic, and biologic risk factors, as well as coronary heart disease (CHD) were applied at the baseline survey according to the manual of the HAPIEE study [21]. All participants from the baseline survey were followed up for all-cause and CVD mortality events until the 31 December 2020, using data from the regional mortality register.

### 2.2. Study Area and Population

The study area was Kaunas—the second largest city in Lithuania with a population of 348,506 in 2006, at the time when the baseline survey started. The population of Kaunas was 156,414 (44.9%) men and 192,092 (55.1%) women. The study object was population of men and women aged 45–72 years. In 2006, 119,098 (156,414 men and 192,092 women) inhabitants of Kaunas city were aged 45–74 years.

### 2.3. Study Sample

Study sample consists of Kaunas men and women (10,980 individuals) aged 45–72 years, stratified by gender and 5-year age groups, randomly selected from the Kaunas population register. Criteria for inclusion in the study: officially declared place of residence and address in the Kaunas city, age from 45 to 72 years, and male or female sex. Criteria for exclusion from the study: place of residence in the city of Kaunas, but officially declared not in the city of Kaunas, age group not between 45 and 72 years old. The response rate at the baseline survey was 65% with 7115 respondents who participated in the survey. 6177 participants (2765 men and 3412 women) were included into the statistical analysis after excluding 923 respondents with missing information on study variables. Ethical approval for HAPIEE study was received from the Kaunas Regional Biomedical Research Ethics Committee (11 January 2005; No. 05/09) and from the Ethics Committee at the University College London. All study participants signed the form of informed consent to participate in the survey and they allowed to use their medical documents during the follow-up.

### 2.4. Sociodemographic, Psychological Well-Being, and Lifestyle Factors

Sociodemographic factors (age and education) were determined using a standard questionnaire at the baseline survey [21]. Age was used as a continuous variable; education was categorized as: 1. Secondary education and lower; 2. College and higher education.

Control Autonomy Self-realization and Pleasure (CASP12) questionnaire was applied for the assessment of PWB; participants were presented with a list of 12 statements which described their lives or how they feel [22]. The answers on a 4-point scale ranged from “never” to “often”, resulting in scores (from 12 to 48). The internal consistency of the CASP-12 scale was evaluated as good (Cronbach’s alpha = 0.74) [23,24]. According to the PWB scores the study participants were categorized: a higher PWB group-included scores equal to median or higher (at baseline survey: >40 in men and >38 in women); a lower PWB group-included scores lower than the median.

A standard questionnaire for evaluation of lifestyle factors and some anthropometric measurements was used. According to the information acquired, smoking status was classified as “never smoking”, “former smoking”, and “current smoking” (individuals who regularly smoked at least 1 cigarette per day). Physical activity was determined by the mean length of time spent per week during leisure time (autumn–winter and spring–summer seasons) doing activities such as gardening, housekeeping, and other physical activities. According to their physical activity during leisure time, the study participants were ranked from the lowest to the highest values and divided into three equal groups (tertiles). The first tertile cut-off (max) is 10 h, which is why this cut-off was used to identify insufficient physical activity.

### 2.5. Objective Measurements

Objective measurements (blood pressure, height, and body weight), and biochemical analyses (high-density lipoprotein (HDL) cholesterol, low-density lipoprotein (LDL) cholesterol, fasting glucose, and triglyceride) were obtained at the baseline survey. Weight and height were determined with a calibrated medical scale. Body mass index (BMI) was calculated: the weight in kilograms divided by the height in meters squared (kg/m^2^). Study participants were divided into four BMI groups: the group with normal weight (BMI 18.5–24.99 kg/m^2^), the group of overweight (BMI 25.0–29.99 kg/m^2^), the group of obese (BMI 30.0–34.9 kg/m^2^), and the group of severe obesity (BMI ≥ 35.0 kg/m^2^). Underweight was defined as BMI < 18.5 kg/m^2^ [25].

Blood pressure (BP) was measured three times with an oscillometric device (Omron M5-1) after at least 5 min of rest in a seated position, and mean values of systolic blood pressure (BP) and diastolic BP were calculated. Arterial hypertension was defined as systolic blood pressure being ≥140 mm Hg and/or diastolic blood pressure being ≥90 mm Hg or self-reporting about a medication prescribed for hypertension during the preceding two weeks.

Fasting blood serum samples were analyzed at the WHO Regional Lipid Reference Centre, Institute of Clinical and Experimental Medicine, Prague (Czech Republic). Serum lipid concentrations (low-density lipoprotein (LDL) cholesterol, triglycerides, and high-density lipoprotein (HDL) cholesterol) were measured on a Roche COBAS MIRA auto-analyzer, using a conventional enzymatic method with reagents from Boehringer-Mannheim Diagnostics and Hoffmann-La Roche. The quality control of biochemistry measures was under the responsibility of the WHO Regional Lipid Reference Centre. Concentration of glucose in capillary blood was determined by the individual glucometer “Glucotrend” [26]. According to the diagnostic criteria for the metabolic syndrome by the Third Report of the National Cholesterol Education Program Adult Treatment Panel III (NCEP-ATP III) definition [27], increased levels of lipids and fasting glucose were defined: HDL cholesterol for men < 1.0 mmol/L, women < 1.3 mmol/L, LDL cholesterol ≥ 3.0 mmol/L, triglycerides ≥ 1.7 mmol/L, fasting glucose level ≥ 6.1 mmol/L.

The criteria for determination of CHD were: (1) a documented history of myocardial infarction (MI) and/or ischemic changes on electrocardiogram (ECG) coded by Minnesota codes (MC) 1–1 or 1–2 [28]; (2) angina pectoris, defined by G. Rose’s questionnaire (without a history of MI and/or MC 1–1 or 1–2) [29]; (3) ischemic changes on ECG coded by MC 1–3, 4–1, 4–2, 4–3, 5–1, 5–2, 5–3, 6–1, 6–2, 7–1, or 8–3 (without MI and/or MC 1–1, 1–2 and without angina pectoris). Stroke was determined according to a documented history of stroke: multiple sources of information (hospital discharge records, records of outpatient departments and other medical documents) were used for evaluation of stroke events. CVD included CHD and/or stroke which were determined at the baseline survey.

### 2.6. The Follow-Up of Mortality Risk from All-Cause and CVD for Endpoints

Study participants were followed for mortality events from 2006 until the 31 December 2020. Mortality data were extracted from the regional mortality register.

The causes of death were grouped as all–causes (A00–Z99: codes of ICD–10) and CVD (I00–I99: codes of ICD–10). During 2006–2020, there were 1339 deaths from any cause (822 in men and 517 in women) and 637 deaths from CVD (387 in men and 250 in women). The mean (SD) duration of the follow-up for the endpoints period was 12.6 (2.79) years.

### 2.7. Statistical Analysis

The data were analyzed using IBM SPSS Statistics (Version 20.0) (IBM Corp. Released 2011. IBM SPSS Statistics for Windows, Version 20.0. Armonk, NY, USA). All the analyses were performed separately for men and women. The distributions of variables were compared in sex groups, using chi-square and z tests, at the baseline survey. Mean differences were tested by applying a *t*-test, *p* < 0.05 values were considered statistically significant. For multivariable analysis all variables that were significantly associated with the risk of mortality over the follow-up in the univariate Cox regression analysis were used. Hazard ratios (HR) and 95% confidence intervals (CI) were calculated for all-cause and CVD mortality in both groups (men and women) using BMI and PWB groups as separate independent factors. Two models were assessed: Model 1 was adjusted for age, education status, arterial hypertension, physical activity and smoking habits, fasting glucose level, triglycerides, HDL cholesterol, and LDL cholesterol; Model 2 was adjusted for all the variables in the Model 1 plus CVD at the baseline survey. To evaluate the combined effect of BMI and PWB on the risk of all-cause and CVD mortality, similar Cox regression models were assessed, and HR and 95% CI were calculated for combined groups; the reference category applied was normal weight plus higher PWB.

## 3. Results

The study participants were followed-up from the beginning of the baseline health examination date until the 31 December 2020. The mean duration and SD of the follow-up of the participants were 12.1 ± 3.26 years among men and 12.9 ± 2.22 years among women. During the follow-up for the endpoints period, the rate of all-cause death in the men group was 24.2% and in the women group was 11.6%, and the rate of death from CVD was 11.1% in the group of men and 5.10% in the group of women. The rates of all-cause death and death from CVD in the men group were two times higher compared to the women group (*p* < 0.001). The mean age (±SD) of respondents at the end of the follow-up in the all-cause mortality group was 69.8 ± 8.4 years for men and 61.7 ± 8.7 years for women; in the CVD mortality group 71.0 ± 7.9 years and 75.3 ± 6.8 years respectively, and in the alive group 69.5.0 ± 7.5 years and 70.0 ± 7.6 years respectively. The mean age of the responders who survived during the follow-up period was significantly lower as compared with responders who died from CVD and from all causes of death (only in the women group).

Table 1 shows the baseline characteristics differentiated by gender. The prevalence of CVD at the baseline survey was higher in women compared to men. Mean age in men and women groups did not differ at the baseline survey, while the prevalence of some of the biological and lifestyle risk factors was different in men and women groups. In the men group, the indicators of arterial hypertension, increased triglycerides level, regular smoking, and physical inactivity levels were higher compared to the women group. However, the prevalence of low level of HDL cholesterol was higher, the mean BMI was higher, and the mean PWB was lower in the group of women compared to men.

Table 2 shows the means of PWB and prevalence of higher and lower PWB in BMI groups according to gender. In the group of men, the mean of PWB was significantly increased in the overweight and obese BMI groups compared to the normal weight BMI group. On the contrary, in the women’s group, the mean of PWB was decreased in obese and severe obese BMI groups compared to the normal weight BMI group. The lower PWB was more prevalent in the underweight, obese, and sever obese BMI group of women compared to the normal weight BMI group.

The risk of all-cause and CVD mortality for BMI and PWB groups according to gender is presented in Table 3. After adjustment for age, education status, smoking habits status, physical activity habits, and biological factors (arterial hypertension, HDL cholesterol, LDL cholesterol, triglycerides, fasting glucose) the overweight (HR = 0.82) was associated with decreased all-cause mortality in the men group, and underweight (HR = 4.08) and severe obesity (HR = 1.40) were associated with increased risk all-cause mortality in women group compared to respondents with normal weight. After an additional adjustment for CVD at the baseline survey, severe obesity (HR = 1.52) significantly increased the risk of mortality from CVD in the men group; on the contrary, the overweight (HR = 0.54) and obesity (HR = 0.62) significantly decreased the mortality from CVD risk in women group compared to responders with normal weight. BMI continuous values (increased per 1 unit) significantly increased the mortality risk from CVD in the men group and all-cause mortality risk in the women group. Lower PWB significantly increased the risk of all-cause and CVD mortality in both groups for men and women compared to the respondents with higher PWB (data adjusted for all variables).

The influence of the combined status of BMI and PWB for the estimation of all-cause and CVD mortality risk is presented in Table 4 (data adjusted for all variables). The combinations of underweight plus lower PWB and severe obesity plus lower PWB increased the all-cause mortality risk in the groups of men (respectively HR = 5.65 and HR = 1.60) and women (respectively HR = 6.02 and HR = 1.77) and increased the risk of mortality from CVD in the group of men (respectively HR = 6.69 and HR = 2.19) compared to respondents with normal weight plus higher PWB. Combination of severe obesity plus higher PWB significantly increased the risk of all-cause and CVD mortality in the men group compared to respondents with normal weight plus higher PWB. Combinations of normal weight plus lower PWB and overweight plus lower PWB significantly increased the risk of all-cause mortality in the group of men. Combination of overweight plus higher PWB decreased CVD mortality risk in the women group.

## 4. Discussion

A notably important result of this study is that during the follow-up period the rates of all-cause death and death from CVD in the group of men were two times higher compared to the group of women; however, the prevalence of CVD at the baseline survey was significantly higher in the women group compared to the men group. The main explanation could be a significant difference in the prevalence of non-communicable risk factors in those groups. Men were 2.49 times more likely to report smoking regularly and 1.7 times more likely to report being physically inactive compared to women at the baseline survey. Meanwhile, women were more educated, and the prevalence of arterial hypertension was less common compared to men. The usual view was that differences in behavior were more important determinants of higher male mortality than inherent sex differences in physiology [30,31]. More recently, a review article analyzing gender differences in CVD highlighted that gender differences cause widespread concerns, and the consideration of gender differences is of great importance for the prevention, diagnosis, treatment, and management of CVD, and the differences are mainly caused by innate genes and environmental influences [32]. However, adjusting for unhealthy behaviors shows that they contribute too but do not fully explain the increased risk of CVD in men [31].

This study raised some more interesting questions. First, why are associations of PWB with BMI and their relationship with mortality risk so little analyzed in the adult population? The research studies showed similar results to our study results when respondents were adults, and the elderly population number is limited [14,15]. Most studies analyzed the association between BMI and PWB, but not their combined impact on mortality risk [8,9]. The association between BMI and PWB frequently is conducted in the children and adolescent populations [33,34]. Moreover, many scientific studies analyzed these two risk factors as increased BMI and lower PWB separately [24,27]. A previous study revealed that lower PWB has been identified as one of the leading risk factors for noncommunicable diseases [9,10] and anthropometric measures such as BMI are a good indicator of all-cause and CVD mortality risk [5,8]. We analyzed risk of all-cause and CVD mortality for PWB and BMI groups separately according to gender, at first. The results obtained in our study confirmed the results from the previous studies indicating that lower PWB significantly increased the risk of all-cause and CVD mortality in men and women compared to respondents with higher PWB (data adjusted for all variables). While consensus has yet to be reached about the optimal level of BMI for health our results suggest that when BMI is above the normal range, for example, severe obesity, in men’s group increased mortality for CVD risk and in women’s group increased all-cause mortality risk. However, on the contrary, in the women group, being overweight decreased the CVD mortality risk, despite, that the data were adjusted for age, education status, lifestyle habits status, biological factors plus CVD at the baseline survey. Previous findings from South Korea showed that overweight or mild obesity is associated with the lowest mortality rate; however, there were no statistically significant findings in the female cohort. Furthermore, the authors of the study suggest that BMI is not a suitable predictor of mortality in women and the current categories of obesity require revision [35]. These results suggest that the BMI cut-off points for observing mortality risk varied depending on gender. The results from a prospective study of a Korean population, showed the reverse J-shaped relationship between BMI and all-cause mortality, with an increased risk among individuals with BMI values in the lower range and it depends on gender [8]. A study in Taipei also demonstrated significant associations between BMI and all-cause and CVD mortality in older age, and the associations were U-shaped, although the risk of all-cause and CVD mortality was significantly lower for both overweight and obesity groups compared to normal BMI between 20-30%, and underweight and severe obesity increased the risk of both total mortality and mortality from circulatory system diseases from 2 to 2.5 times [4].

The scientific results show that being obese is frequently related to deteriorated psychological well-being (PWB) and quality of life, in comparison with normal-weight individuals [15,16]. The results from our data also showed that the combined status of BMI and PWB plays an important role in all-cause and CVD mortality risk. We found that severe obesity, independent of on the lower or higher PWB, increased the risk of all-cause mortality in the groups of men and women (respectively HR = 1.60 and HR = 1.77). Lower PWB combined with underweight increased the risk of all-cause mortality in men and women groups more than five times (respectively HR = 5.65 and HR = 6.02), and the risk of CVD in men group, more than 6 times (HR = 6.69). In the group of men, despite the weight being normal, lower PWB increased the risk of all-cause mortality 1.81 times compared to responders with normal weight plus higher PWB. Thus, we did not find significant associations between the underweight plus higher PWB and all-cause and CVD mortality risk in men and women groups. Previous studies revealed that changes in PWB among older dwellers were associated with the quality of social contacts, being a member of a social organization; retirement negatively affected the PWB of older women [13,24,36].

Second question, why we did not find the associations between severe obesity plus lower PWB, and the combination of underweight weight plus lower PWB with an increased risk of mortality from CVD in the women group? One of the reasons could be too small number of deaths from CVD in the group of women compared to men. Several previous studies found sex differences in the association between PWB and CVD [9,37] and sex differences in the association between BMI and CVD [38]. This might be contributed by different biological risk factors and by the different prevalence of unhealthy lifestyle habits (such as regular smoking, and physical inactivity) in men and women groups. Another reason, raised by researchers from Portugal is that relation between BMI and PWB may not be linear across all outcomes, with unique patterns emerging for the association of obesity level with specific dimensions of PWB [39].

### 4.1. Strengths

The main strengths of the present study: prospective design, large sample size, and a wide interval of age of the study participants (middle-aged and elderly 45–72 year old individuals at baseline). Data collection using standardized and validated study methods, long follow-up period (from 2006–2008 to the 31 December 2020), many potential confounders included into statistical analyses (10 variables in fully adjusted multivariate regression models data) add additional strengths to the study. Furthermore, measurement of BMI and evaluation of PWB are simple and universally available, which makes the study results easily applicable to clinical practice.

### 4.2. Limitations

The present study also has some limitations. First, despite the adjustment for multiple confounders, we did not include the family history of CVD and dietary habits in our statistical models which may have influenced the results of the study. Second, this study relied on some self-reported results (PWB and lifestyle behaviors) which might be affected by recall bias, and this could have resulted in overestimation or underestimation of the determined outcomes. Third, information about the study population during the follow-up period concerning diseases or additional health disorders, emerging new risk factors for chronic diseases and harmful lifestyles, their duration, was unavailable.

## 5. Conclusions

The combination of severe obesity, independently on lower or higher PWB, and the combination of underweight plus lower PWB, are strong predictors for all-cause and CVD mortality risk in men and strong predictors for all-cause mortality risk in women. The lower PWB status, despite normal or overweight BMI, significantly increased all-cause mortality risk in men. However, the combination of overweight plus higher PWB decreased CVD mortality risk in the women group. Based on our results, we can provide a recommendation: if a person is diagnosed with severe obesity or underweight by a doctor, we recommend evaluating the PWB for this person; if the determined PWB is higher, the mortality risk is not as high compared to the case if the PWB is lower.

## Figures and Tables

**Table 1 medicina-58-01591-t001:** Baseline characteristics of men and women of the Kaunas HAPIEE study (2006–2008).

	MEN	WOMEN	*p*
Variables	n = 2765	n = 3412	
Age, years, mean ± SD	57.5 ± 7.96	57.3 ± 7.85	>0.05
Education, %			
Secondary and lower	46.8	37.1	<0.001
College and higher	53.2	62.9	
HDL cholesterol,			
Men < 1.0 mmol/L, women < 1.3 mmol/L, %	11.9	28.3	<0.001
LDL cholesterol ≥ 3.0 mmol/L, %	75.1	78.6	0.001
Triglycerides ≥ 1.7 mmol/L, %	29.0	25.2	0.001
Fasting glucose ≥ 6.1 mmol/L, %	30.9	30.7	>0.05
Arterial hypertension %	76.9	67.4	<0.001
Body mass index, mean ± SD	28.4 ± 4.54	29.6 ± 5.69	<0.001
Body mass index, %			<0.001
<18.5 kg/m^2^	0.2	0.3	
18.5–24.9 kg/m^2^	22.2	22.0	
25.0–29.9 kg/m^2^	44.7	36.0	
30.0–34.9 kg/m^2^	25.0	24.9	
≥35.0 kg/m^2^	7.9	16.9	
PWB, mean ± SD	39.3 ± 5.47	37.9 ± 6.05	<0.001
PWB groups			0.004
Higher	53.2	56.9	
Lower	46.8	43.1	
Regular smoking, %	34.1	13.7	<0.001
Physical inactivity, %	33.0	19.0	<0.001
Prevalence of CVD at baseline survey, %	17.2	19.8	0.011

Chi square test for distributions and *t*-test for means were used to compare differences of variables between sex groups. Data were adjusted by age. CVD—cardiovascular diseases; HAPIEE—Health, Alcohol and Psychosocial factors In Eastern Europe; HDL—high-density lipoprotein; LDL—low-density lipoprotein, PWB—psychological well-being; SD—standard deviation.

**Table 2 medicina-58-01591-t002:** Prevalence of higher and lower PWB in BMI groups according to gender.

		BMI	Groups			
	Underweight	Normal Weight	Overweight	Obesity	Severe Obesity	
	<18.5 kg/m^2^	18.5–24.9 kg/m^2^	25.0–29.9 kg/m^2^	30.0–34.9 kg/m^2^	≥35.0 kg/m^2^	*p*
**MEN**						
PWB, mean (SD)	38.5 (6.31)	38.7 (5.62)	39.6 (5.31) *	39.5 (5.53) *	38.6 (5.59)	0.002
PWB Higher, %	54.5	49.0	55.3	54.2	49.3	>0.05
PWB Lower, %	45.5	51.0	44.7	45.8	50.7	>0.05
**WOMEN**						
PWB, mean (SD)	37.7 (4.24)	38.3 (5.96)	38.2 (5.73)	37.4 (6.34) *	37.0 (6.43) *	<0.0001
PWB Higher, %	50.0 *	60.3	59.0	53.1*	52.6 *	0.002
PWB Lower, %	50.0 *	39.7	41.0	46.9 *	47.4 *	0.002

Chi square and Z tests for distributions and F-test (ANOVA) for means were used to compare differences of PWB between BMI groups; for multiple comparison Bonferroni correction was used. * *p* < 0.05 compared with the normal weight BMI group (18.5–24.9 kg/m^2^).

**Table 3 medicina-58-01591-t003:** All-cause and CVD mortality risk for BMI and PWB groups according to gender (follow–up for endpoints period mean (SD) 12.6 (2.79) years).

		MEN					WOMEN	
	All-Cause	Mortality	CVD	Mortality	All-Cause	Mortality	CVD	Mortality
	HR *	95% CI	HR **	95% CI	HR *	95% CI	HR **	95% CI
**BMI groups**								
<18.5 kg/m^2^	2.32	0.95–5.68	2.85	0.70–11.7	**4.08**	1.63–10.2	1.62	0.38–6.85
18.5–24.9 kg/m^2^	**1**		**1**		**1**		**1**	
25.0–29.9 kg/m^2^	**0.82**	0.68−0.97	0.90	0.69−1.19	0.93	0.70−1.24	**0.54**	0.36−0.80
30.0–34.9 kg/m^2^	0.85	0.69−1.04	1.03	0.76−1.39	1.06	0.79−1.42	**0.62**	0.41−0.92
≥35.0 kg/m^2^	1.09	0.84−1.41	**1.52**	1.06−2.17	**1.40**	1.03−1.89	0.94	0.63−1.43
**BMI ***,** kg/m^2^	1.00	0.99−1.02	**1.03**	1.01−1.05	**1.03**	1.01−1.04	1.02	0.99−1.04
**PWB groups**								
Higher	**1**		**1**		**1**		**1**	
Lower	**1.40**	1.22−1.61	**1.29**	1.05−1.58	**1.30**	1.09−1.55	**1.48**	1.14−1.91
**PWB *****	**0.96**	0.95−0.97	**0.97**	0.96−0.99	**0.98**	0.97−0.99	**0.97**	0.95−0.99

Cox regression analysis was used. **HR *** (hazard ratios) adjusted for age, education status, smoking habits status, physical activity habits, and biological factors (arterial hypertension, HDL cholesterol, LDL cholesterol, triglycerides, fasting glucose). **HR **** adjusted for all the variables included in HR * for all-cause mortality plus cardiovascular disease (CVD) at baseline survey. ******* BMI and PWB continuous variables (increased per 1 unit). **BMI**—body mass index; **PWB**—psychological well-being; CI—confidence interval.

**Table 4 medicina-58-01591-t004:** The influence of combined status of body mass index and psychological well-being for the estimation of all-cause mortality risk and CVD mortality risk (follow-up for endpoints period mean (SD) 12.6 (2.79) years).

		MEN					WOMEN	
BMI and PWB Groups	All-Cause	Mortality	CVD	Mortality	All-Cause	Mortality	CVD	Mortality
	HR *	95% CI	HR **	95% CI	HR *	95% CI	HR **	95% CI
Normal weight + higher PWB	**1**		**1**		**1**		**1**	
Underweight + higher PWB	1.11	0.15−8.05	−	−	2.05	0.28−15.2	2.88	0.37−22.7
Overweight + higher PWB	0.98	0.73−1.29	1.04	0.67−1.60	0.97	0.63−1.50	**0.52**	0.27−1.00
Obesity + higher PWB	1.06	0.78−1.45	1.24	0.78−1.97	1.11	0.71−1.73	0.56	0.29−1.10
Severe obesity+ higher PWB	**1.52**	1.00−2.31	**1.84**	1.02−3.32	1.57	0.99−2.47	1.17	0.62−2.22
Underweight + lower PWB	**5.65**	2.05−15.6	**6.69**	1.59−28.2	**6.02**	1.80−20.1	1.69	0.22−13.2
Normal weight + lower PWB	**1.81**	1.35−2.42	1.46	0.91−2.33	1.43	0.89−2.33	1.74	0.92−3.31
Overweight+ lower PWB	**1.35**	1.02−1.79	1.39	0.90−2.15	1.33	0.87−2.05	0.92	0.50−1.69
Obesity+ lower PWB	1.27	0.93−1.72	1.30	0.82−2.06	1.46	0.94−2.25	0.94	0.51−1.74
Severe obesity+ lower PWB	**1.60**	1.12−2.30	**2.19**	1.33−3.63	**1.77**	1.13−2.75	1.21	0.65−2.26

Cox regression analysis was used. **HR *** (hazard ratios) adjusted for age, education status, smoking habits status, physical activity habits, and biological factors (arterial hypertension, HDL cholesterol, LDL cholesterol, triglycerides, fasting glucose). **HR **** adjusted for all the variables included in HR * for all-cause mortality plus cardiovascular disease (CVD) status at baseline survey. **BMI**—body mass index; **PWB**—psychological well-being; **CI**—confidence interval.

## Data Availability

The datasets used and/or analyzed during the current study are available from the corresponding author on reasonable request.

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
