# Peer review of "The Prognostic Value of Combined Status of Body Mass Index and Psychological Well-Being for the Estimation of All-Cause and CVD Mortality Risk: Results from a Long-Term Cohort Study in Lithuania"

_medicina, 2022, doi:10.3390/medicina58111591_

Round 1

Reviewer 1 Report

Thank you very much for the opportunity to review this interesting piece of research. The study is of high quality. I have one question only.

1.      The finding that obesity, underweight, and mental health problems are the major independent risk factors is not new. However, it is not well understood what the added value of combining these risk factors in one index is. Did you find that these factors have interaction or that one is an effect modifier for another? If not, what is the rationale for combining them? Based on your results, would you rather recommend combining them or not?

Author Response

Dear Reviewer

We would like to thank you for your very valuable comments and new experiences regarding our manuscript “The prognostic value of combined status of body mass index and psychological well-being for the estimation of all-cause and CVD mortality risk: results from a long-term cohort study in Lithuania” (medicina-1968251). We have revised the manuscript according to the reviewers' comments We hope that the quality of the revised version of our manuscript has been improved as suggested.

I sincerely thank you on behalf of all the co-authors.

The answers to Reviewer 1

Questions. The finding that obesity, underweight, and mental health problems are the major independent risk factors is not new. However, it is not well understood what the added value of combining these risk factors in one index is. Did you find that these factors have interaction or that one is an effect modifier for another? If not, what is the rationale for combining them? Based on your results, would you rather recommend combining them or not?

Answers: We added additional information in the Introduction section and Conclusion section.

“The Encyclopedia of Gerontology and Population Aging emphasizes that obesity can affect emotional well-being through psychological, physical, and social factors related to social acceptance and self-perception [14]. Indeed, the scientific results show that being obese is frequently related to deteriorated psychological well-being (PWB) and quality of life, in comparison with normal-weight individuals [15,16]. Previous our studies analyzed these two risk factors separately. According to this scientific literature, we hypothesized that combinations of increased body mass index (BMI) level and low PWB are strongly positively associated with the risk of mortality and have multiplicative interaction.”

“Based on our results, we can provide a recommendation: if a person is diagnosed with severe obesity or underweight by a doctor, we recommend evaluating the PWB for this person; if the determined PWB is higher, the mortality risk is not as high compared to the case if the PWB is lower.”

References:

  1. von Humboldt, S. Obesity, Perceived Weight Discrimination, and Well-Being. In Encyclopedia of Gerontology and Population Aging. Gu, D.; Dupre, M. Eds.; Springer Nature, Switzerland, 2019, https://doi.org/10.1007/978-3-319-69892-2_80-1
  2. Giuli, C.; Papa, R.; Marcellini, F.; Boscaro, M.; Faloia, E.; Lattanzio, F.; Tirabassi, G.; Bevilacqua, R. The role of psychological well-being in obese and overweight older adults. Int. Psychogeriatr. 2016, 28, 171-172, doi: 10.1017/S1041610215001313.
  3. Kalish, V.B. Obesity in older adults. Prim. Care. 2016, 43, 137–144, https://doi.org/10.1016/j.pop.2015.10.002

Reviewer 2 Report

In my opinion the paper was informative and will provide a valuable source document for anyone requiring a primer to know and understand this issue. Some changes are needed:    

  • Line 19. State the study design applied in this research.
  • Lines 45-46: Both references (No 9 and No 10) refer to the population of Lithuania, add references that give results from studies conducted in different countries.  
  • Lines 48-50: This sentence not precisely enough for the stated claim `both developed and developing countries` cites 2 references (No 11 and No 12), which both refer to results of research conducted in the Italian population. It is necessary to cite appropriate references for the given claim.
  • Line 60: Add a new subsection `Study design`, with all necessary information.  
  • Line 62: Add a new subsection `Study population`, with a detailed description. 
  • Lines 63-68: State the criteria for inclusion and exclusion in the study. 
  • Lines 85-89: Cite the reference that has published results of the assessment of psychometric characteristics of the Control Autonomy Self-realization and Pleasure (CASP12) questionnaire in the population of Lithuania.  
  • Lines 119-125: State whether at the `at baseline` the value of Hemoglobin A1c was determined. 
  • Line 131-132: For `The previous stroke`, explain about which `document` it refers to in the study. 
  • Lines 142-156: State whether collinearity of variables was determined, using what method, what were the results and how were they addressed as a potential issue. 
  • Lines 158-160: Table 1, that presents `Baseline characteristics of men and women at the baseline survey of the Kaunas HAPIEE 169 study (2006-2008).` shows at the end of Table 1 and 2 variables that refer to `Dead from all-causes of death, %` and `Dead from CVD, %`. Explain why these two variables are shown in Table 1.  
  • Line 159: If Table 1 presents `Baseline characteristics of men and women at the baseline survey of the Kaunas HAPIEE 169 study (2006-2008).`, explain why Table 1 has variable `Follow-up, mean ± SD`. In the description of this results provide an explanation for significant difference by sex for the length of follow-up period. 
  • Line 171: State which statistical test was used to assess the differences in variables presented in Table 1. 
  • Line 175: Add a new Table that will present the basic demographic characteristics of persons at the end of the study (age and sex are of course in the database of the used registry), education, but surely for data of diabetes, arterial hypertension too, in comparison with persons that have survivd at the end of the study period.   
  • Line 183: State which statistical test was used to assess the differences in variables presented in Table 2. 
  • Line 199: State which statistical test was used to assess the differences in variables presented in Table 3.
  • Line 221: State which statistical test was used to assess the differences in variables presented in Table 4.
  • Line 227: A notably important result of this study is the two-times higher `Dead from all-causes of death, %` and `Dead from CVD, %` in males than in females at the end of the study, with a significantly higher `Prevalence of CVD at baseline survey, %` in females in comparison to males. Discuss this and provide possible explanations for these results. 
  • Lines 230-231: Cite appropriate references for the mentioned `studies studies showed similar results to`.  
  • Lines 246-248: Apart from the given statement in this sentence, which presents repetition of presented results, it is necessary to provide a possible explanation for such results, i.e. differences by sex. 
  • Lines 248-249: Since the Body Mass Index in Table 1 was shown as a continuous variable, state whether the observed differences in mortality by sex have remained when analysing Body Mass Index as a continuous variable.  
  • Lines 258-276: Reconstruct this paragraph, because it contains redundant repetition from previous sections. Results of this study should be compared to results of similar studies in the world, it is not enough to just compare them with the results in the same country - Lithuania and in the same study to which it seems that the cited ref. No. 24 refers to.
  • Lines 280-290: Some of the stated questions could have been resolved either while planning the study or during the statistical analysis of data that are presented in this manuscript. 
  • Lines 282-283: In the context of this sentence, discuss the statistically significantly longer `Follow-up, mean ± SD` for females in comparison to males.  
  • Lines 294-295: It is necessary to define in the section `Materials and Methods` in a new subsection `Study design` which `standardized and validated study methods` was applied in this manuscript.  

Author Response

Dear Reviewer  

We would like to thank you for your very valuable comments and new experiences regarding our manuscript “The prognostic value of combined status of body mass index and psychological well-being for the estimation of all-cause and CVD mortality risk: results from a long-term cohort study in Lithuania” (Manuscript Submission medicina-1968251). We have revised the manuscript according to the reviewers' comments We hope that the quality of the revised version of our manuscript has been improved as suggested.

I sincerely thank you on behalf of all the co-authors.

The answers to Reviewer 2

  • Line 19. State the study design applied in this research.

Answer: We corrected the sentence. “Initial data were collected within the framework of the international cohort project HAPIEE study from 2006 to 2008.’

  • Lines 45-46: Both references (No 9 and No 10) refer to the population of Lithuania, add references that give results from studies conducted in different countries.

Answer: We added additional references that present results from studies conducted in different countries.

References:

  1. Ortega, F.B.; Lee, D.C.; Sui, X; Kubzansky, L.D., Ruiz, J.R.; Baruth, M.; Castillo, M.J.; Blair, S.N. Psychological wellbeing, cardiorespiratory fitness, and long-term survival. Am. J. Prev. Med. 2010, 39, 440-448, doi: 10.1016/j.amepre.2010.07.015.
  2. Stein, D.J.; Benjet, C.; Gureje, O.; Lund, C.; Scott, K.M.; Poznyak, V.; van Ommeren, M. Integrating mental health with other non-communicable diseases. BMJ 2019, 364, doi: https://doi.org/10.1136/bmj.l295
  3. Trudel-Fitzgerald, C.; Kubzansky, L. D.; VanderWeele, T. J. A review of psychological well-being and mortality risk: Are all dimensions of psychological well-being equal? In Measuring well-being: Interdisciplinary perspectives from the social sciences and the humanities. Lee M.T.; Kubzansky, L.D.; VanderWeele T.J., Eds.; Oxford University Press, New York, 2021, pp.136-187. https://doi.org/10.1093/oso/9780197512531.003.0006

  • Lines 48-50: This sentence not precisely enough for the stated claim `both developed and developing countries` cites 2 references (No 11 and No 12), which both refer to results of research conducted in the Italian population. It is necessary to cite appropriate references for the given claim.

Answer: We added additional references that present results from studies conducted in different countries.

References:

  1. Dinsa, G.D.; Goryakin, Y.; Fumagalli, E.; Suhrcke, M. Obesity and socioeconomic status in developing countries: a systematic review. Obes. Rev. 2012, 13, 1067-1079, doi: 10.1111/j.1467-789X.2012.01017.x
  2. von Humboldt, S. Obesity, Perceived Weight Discrimination, and Well-Being. In Encyclopedia of Gerontology and Population Aging. Gu, D.; Dupre, M. Eds.; Springer Nature, Switzerland, 2019, https://doi.org/10.1007/978-3-319-69892-2_80-1

  • Line 60: Add a new subsection `Study design`, with all necessary information.

Answer: We added a new subsection 2.1. “Study design”.

  • Line 62: Add a new subsection `Study population`, with a detailed description.

Answer: We added a new subsection 2.2. “Study area and population”.

  • Lines 63-68: State the criteria for inclusion and exclusion in the study.

Answer: We added the description of the criteria for inclusion and exclusion in the study in the subsection 2.3. “Study sample”.

  • Lines 85-89: Cite the reference that has published results of the assessment of psychometric characteristics of the Control Autonomy Self-realization and Pleasure (CASP12) questionnaire in the population of Lithuania.

Answer: We added additional references:

References:

  1. Sapranaviciute-Zabazlajeva, L. Psychological well-being and the risk of cardiovascular diseases in middle aged and elderly population. Doctoral dissertation (R1). Field of Science: Public health (M004), 2015, 211 p, https://www.lsmuni.lt/cris/handle/20.500.12512/16180 (in Lithuanian language)
  2. Sapranaviciute-Zabazlajeva, L.; Luksiene, D.; Virviciute, D.; Kranciukaite-Butylkiniene, D.; Bobak, M.; Tamosiunas, A. Changes in psychological well-being among older Lithuanian city dwellers: Results from a cohort study. Int. J. Clin. Health. Psychol. 2018, 18, 218-226. doi: 10.1016/j.ijchp.2018.05.002.

  • Lines 119-125: State whether at the `at baseline` the value of Hemoglobin A1c was determined.

Answer: Hemoglobin A1c was not determined at the baseline.

  • Line 131-132: For `The previous stroke`, explain about which `document` it refers to in the study.

Answer: Data on previous stroke patients were obtained from the regional population-based Stroke Register which was conducted according to the WHO MONICA project protocol. Multiple sources of information (hospital discharge records, records of outpatient departments and other medical documents) were used for evaluation of stroke events.

We added additional information in subsection “2.5. Objective measurements”

  • Lines 142-156: State whether collinearity of variables was determined, using what method, what were the results and how were they addressed as a potential issue.

Answer: We performed the collinearity test and calculated the non-parametric Spearman correlation coefficient, very weak correlations (r<0.3) were found between the variables. In this way, the collinearity problem was not detected.

  • Lines 158-160: Table 1, that presents `Baseline characteristics of men and women at the baseline survey of the Kaunas HAPIEE 169 study (2006-2008).` shows at the end of Table 1 and 2 variables that refer to `Dead from all-causes of death, %` and `Dead from CVD, %`. Explain why these two variables are shown in Table 1.

Answer: We revised Table 1 and added information about the follow-up for the endpoints period and the rates of all-cause death and death from CVD in the men and women groups in Results section.

„ The study participants were followed-up from the beginning of the baseline health examination date until the 31st of December 2020. The mean duration and SD of the follow-up of the participants were 12.1±3,26 years among men and 12.9±2.22 years among women. During the follow-up for the endpoints period, the rate of all-cause death in the men group was 24.2% and in the women group was 11.6%, and the rate of death from CVD was 11.1% in the group of men and 5.10% in the group of women. The rates of all-cause death and death from CVD in the men group were two times higher compared to the women group (p<0.001). “

  • Line 159: If Table 1 presents `Baseline characteristics of men and women at the baseline survey of the Kaunas HAPIEE 169 study (2006-2008).`, explain why Table 1 has variable `Follow-up, mean ± SD`. In the description of this results provide an explanation for significant difference by sex for the length of follow-up period.

Answer: The mean follow-up years significantly differ between men and women because more men died earlier during the follow-up period (until 31 December 2020) and their mean follow-up period decreased compared to women. Meanwhile, most women survived longer during the follow-up period.

  • Line 171: State which statistical test was used to assess the differences in variables presented in Table 1.

Answer: We have provided information about statistical tests in Table 1: Chi square test for distributions and T-test for means were used to compare differences of variables between sex groups.

  • Line 175: Add a new Table that will present the basic demographic characteristics of persons at the end of the study (age and sex are of course in the database of the used registry), education, but surely for data of diabetes, arterial hypertension too, in comparison with persons that have survived at the end of the study period.

Answer:

Sorry, we cannot add a new Table that will present the basic demographic characteristics of persons at the end of the study. The main reason is that we do have no special data about our responders at the end of follow-up (data on diabetes, arterial hypertension, etc.). The cohort study's main aim was to evaluate the data about endpoints (death events) using data on mortality extracted from the regional mortality register. 

We added additional information to the Results section about the mean age of men and women in the all-cause mortality group, CVD mortality group and Alive group.

“The mean age (±SD) of respondents at the end of the follow-up in the all-cause mortality group was 69.8±8.4 years for men and 61.7±8.7 years for women; in the CVD mortality group 71.0±7.9 years and 75.3±6.8 years respectively, and in the alive group 69.5.0±7.5 years and 70.0±7.6 years respectively. The mean age of the responders who survived during the follow-up period was significantly lower as compared with responders who died from CVD and from all causes of death (only in the women group).”

  • Line 183: State which statistical test was used to assess the differences in variables presented in Table 2.

Answer: We have provided information about statistical tests in Table 2: Chi square and Z tests for distributions and F- test (ANOVA) for means were used to compare differences of PWB between BMI groups; for multiple comparison Bonferroni correction was used.

  • Line 199: State which statistical test was used to assess the differences in variables presented in Table 3.

Answer: We have provided information about statistical tests in Table 3: Cox regression analysis was used.

  • Line 221: State which statistical test was used to assess the differences in variables presented in Table 4.

Answer: We have provided information about statistical tests in Table 4: Cox regression analysis was used.

  • Line 227: A notably important result of this study is the two-times higher `Dead from all-causes of death, %` and `Dead from CVD, %` in males than in females at the end of the study, with a significantly higher `Prevalence of CVD at baseline survey, %` in females in comparison to males. Discuss this and provide possible explanations for these results.

Answer: We have provided possible explanation in Discussion section.

“A notably important result of this study is that during the follow-up period the rates of all-cause death and death from CVD in the group of men were two times high-er compared to the group of women, however, the prevalence of CVD at the baseline survey was significantly higher in the women group compared to the men group. The main explanation could be a significant difference in the prevalence of non-communicable risk factors in those groups. Men were 2.49 times more likely to report smoking regularly and 1.7 times more likely to report being physically inactive com-pared to women at the baseline survey. Meanwhile, women were more educated, and the prevalence of arterial hypertension was less common compared to men. The usual view was that differences in behavior were more important determinants of higher male mortality than inherent sex differences in physiology [30,31]. More recently, a re-view article analyzing gender differences in CVD, highlighted that gender differences cause widespread concerns, and the consideration of gender differences is of great im-portance for the prevention, diagnosis, treatment, and management of CVD, and the differences are mainly caused by innate genes and environmental influences [32]. However, adjusting for unhealthy behaviors shows that they contribute too but do not fully explain the increased risk of CVD in men [31].”

References:

  1. Barrett-Connor, E. Sex differences in coronary heart disease. Why are women so superior? Circulation. 1997, 95, 252-264, doi: 10.1161/01.cir.95.1.252.
  2. Möller-Leimkühler, A.M. Gender differences in cardiovascular disease and comorbid depression. Dialogues in Clinical Neuroscience. 2007, 9, 71-83, DOI: 10.31887/DCNS.2007.9.1/ammoeller
  3. Gao, Z.; Chen, Z.; Sun, A.; Deng, X.; Gender differences in cardiovascular disease. Medicine in Novel Technology and Devices. 2019, 4, https://doi.org/10.1016/j.medntd.2019.100025

  • Lines 230-231: Cite appropriate references for the mentioned `studies studies showed similar results to.

Answer: We added additional references that present similar results.

References:

  1. von Humboldt, S. Obesity, Perceived Weight Discrimination, and Well-Being. In Encyclopedia of Gerontology and Population Aging. Gu, D.; Dupre, M. Eds.; Springer Nature, Switzerland, 2019, https://doi.org/10.1007/978-3-319-69892-2_80-1
  2. Giuli, C.; Papa, R.; Marcellini, F.; Boscaro, M.; Faloia, E.; Lattanzio, F.; Tirabassi, G.; Bevilacqua, R. The role of psychological well-being in obese and overweight older adults. Int. Psychogeriatr. 2016, 28, 171-172, doi: 10.1017/S1041610215001313.

  • Lines 246-248: Apart from the given statement in this sentence, which presents repetition of presented results, it is necessary to provide a possible explanation for such results, i.e. differences by sex.

Answer: We modified the Discussion section and added additional reference.

“Previous findings from South Korea showed that overweight or mild obesity is associated with the lowest mortality rate, however, there were no statistically significant findings in the female cohort. Furthermore, the authors of the study suggest that BMI is not a suitable predictor of mortality in women and the current categories of obesity re-quire revision. [36]”.

Reference:

  1. Yu, S.Y.; Kim, B.S.; Won, C.W.; Choi, H.; Kim, S.; Kim, H.W.; Kim, M.J. Body Mass Index and Mortality according to Gender in a Community-Dwelling Elderly Population: The 3-Year Follow-up Findings from the Living Profiles of Older People Surveys in Korea. Korean J. Fam. Med. 2016, 37, 317-322, doi: 10.4082/kjfm.2016.37.6.317.

  • Lines 248-249: Since the Body Mass Index in Table 1 was shown as a continuous variable, state whether the observed differences in mortality by sex have remained when analysing Body Mass Index as a continuous variable.

Answer: BMI continuous values (increased per 1 unit) significantly increased the mortality risk from CVD in the men group and all-cause mortality risk in the women group. These data are presented in Table 3. We added this information to the Results section.

  • Lines 258-276: Reconstruct this paragraph, because it contains redundant repetition from previous sections. Results of this study should be compared to results of similar studies in the world, it is not enough to just compare them with the results in the same country - Lithuania and in the same study to which it seems that the cited ref. No. 24 refers to.

Answer: We reconstructed this paragraph, also added additional references that present results from studies conducted in different countries.

References:

  1. Gyasi, R.; Phillips, D.; Abass, K. Social support networks and psychological wellbeing in community-dwelling older Ghanaian cohorts. International Psychogeriatrics. 2019, 31, 1047-1057, doi:10.1017/S1041610218001539
  2. Trudel-Fitzgerald, C.; Kubzansky, L. D.; VanderWeele, T. J. A review of psychological well-being and mortality risk: Are all dimensions of psychological well-being equal? In Measuring well-being: Interdisciplinary perspectives from the social sciences and the humanities. Lee M.T.; Kubzansky, L.D.; VanderWeele T.J., Eds.; Oxford University Press, New York, 2021, pp.136-187. https://doi.org/10.1093/oso/9780197512531.003.0006

  • Lines 280-290: Some of the stated questions could have been resolved either while planning the study or during the statistical analysis of data that are presented in this manuscript.

Answer: We agree with your remark that some of the stated questions could have been resolved either while planning the study. We have no possibility to plan the study because the Kaunas centre started from wave 2 of the HAPIEE study when the protocol and design were already approved.

  • Lines 282-283: In the context of this sentence, discuss the statistically significantly longer `Follow-up, mean ± SD` for females in comparison to males.

Answer: We added additional information associations of gender and CVD mortality risk at the beginning of the Discussion section.

The mean follow-up years significantly differ between men and women because more men died earlier during the follow-up period (until 31 December 2020) and their mean follow-up period decreased compared to women. Meanwhile, most women survived longer during the follow-up period.

  • Lines 294-295: It is necessary to define in the section `Materials and Methods` in a new subsection `Study design` which `standardized and validated study methods` was applied in this manuscript.

Answer: We added in the new subsection “Study design” an information which standardized and validated study methods were applied.

Round 2

Reviewer 2 Report

I would like to thank the authors for carefully revising their manuscript. The authors have satisfactorily responded to all my questions and made the necessary changes to the manuscript. I believe that the changes they have made have significantly improved the manuscript.